# Impact of Sarcopenia on Clinical Outcomes in a Cohort of Caucasian Active Crohn’s Disease Patients Undergoing Multidetector CT-Enterography

**DOI:** 10.3390/nu14173460

**Published:** 2022-08-23

**Authors:** Olga Maria Nardone, Andrea Ponsiglione, Roberto de Sire, Giulio Calabrese, Raffaele Liuzzi, Anna Testa, Alessia Dalila Guarino, Oriana Olmo, Antonio Rispo, Luigi Camera, Fabiana Castiglione

**Affiliations:** 1Gastroenterology, Department of Public Health, University of Naples Federico II, 80131 Naples, Italy; 2Department of Advanced Biomedical Sciences—Section of Diagnostic Imaging, University Federico II of Naples, 80131 Naples, Italy; 3Gastroenterology, Department of Clinical Medicine and Surgery, University Federico II of Naples, 80131 Naples, Italy; 4Institute of Biostructures and Bioimaging (National Research Council), University “Federico II”, 80131 Naples, Italy

**Keywords:** Crohn’s disease, sarcopenia, malnutrition, CT-enterography

## Abstract

(1) Background: Sarcopenia has a high incidence in Crohn’s disease (CD) with considerable heterogeneity among ethnicities and variable impact on clinical outcomes. Aim: to assess the impact of sarcopenia on clinical outcomes in a cohort of Caucasian patients with active CD undergoing CT-enterography (CTE) for clinical assessment. We further investigated the prevalence of sarcopenia and its predictors. (2) Methods: Caucasian CD patients with moderate–severe clinical activity, who underwent CTE in an emergency setting, were retrospectively recruited. The skeletal muscle index (SMI) at the third lumbar vertebra was used to detect sarcopenia in the early stages. Clinical malnutrition was defined according to global clinical nutrition criteria. Clinical outcomes included the rate of surgery and infections within one year. (3) Results: A total of 63 CD patients (34 M; aged 44 ± 17 years) were recruited, and 48 patients (68.3%) were sarcopenic. Malnutrition occurred in 28 patients (44.4%) with a significant correlation between body mass index (BMI) and sarcopenia (r = 0.5, *p* < 0.001). The overall rate of surgery was 33%, without a significant difference between sarcopenic and non-sarcopenic (*p* = 0.41). The rate of infection in patients with sarcopenia was significantly higher than in non-sarcopenic (42%vs15%, *p* = 0.03). BMI (OR 0.73,95%, CI 0.57–0.93) and extraintestinal manifestations (EIM) (OR 19.2 95%, CI 1.05–349.1) were predictive of sarcopenia (*p* < 0.05). (4) Conclusions: Sarcopenia was associated with an increased rate of infections, and it was observed in 68.3% of the Caucasian cohort with active CD.

## 1. Introduction

The prevalence of inflammatory bowel disease (IBD), including Crohn’s disease (CD) and ulcerative colitis (UC), is increasing significantly worldwide and has a huge impact on quality of life, social functioning, and psychological health [1,2,3].

Emerging data suggest that the assessment of body composition is important in patients with IBD, especially for CD, and may help to identify those at higher risk for either disease or treatment-related complications [4,5,6,7].

Indeed, patients with IBD are at risk of malnutrition due to an imbalance between nutritional requirement and caloric loss, especially in the active disease state [8,9]. Closely related to malnutrition is the concept of sarcopenia, a result of chronic inflammation and gut dysbiosis due to IBD [10,11,12,13].

The European Working Group on Sarcopenia in Older People defined sarcopenia as a progressive and generalized muscle failure, characterized by loss of either muscle mass or strength, with physical performance impairment [14]. It represents a major health burden in industrialized countries leading to physical disability, reducing the quality of life, and increasing the risk of hospital admissions and mortality [15]. A recent meta-analysis reported that sarcopenia was described in more than one-third of patients who suffered from IBD [16], and thus it has increasingly become part of clinical management for patients with IBD.

Sarcopenia and body composition can be evaluated by radiological imaging techniques at the level of the third lumbar vertebra, such as computed tomography (CT) and magnetic resonance imaging (MRI) [17,18,19,20]. These high-resolution imaging systems are reliable and easy tools to quantify muscle mass and detect sarcopenia in its early stages.

However, there is considerable heterogeneity in the assessment of sarcopenia and body composition parameters since its cut-off values vary among the studies and ethnicity, and there are differences in body size between Western and Asian patients with IBD. In addition, it is noteworthy that sarcopenia represents a poor prognostic factor in IBD patients [21,22,23,24]. Several studies have shown that skeletal muscle index strongly correlates with clinical outcomes such as intestinal resection and postoperative complications in CD patients, and visceral adipose tissue/height index is associated with exacerbation of CD [5,16,21,22,23,24]. Hence, a comprehensive assessment of the sarcopenia and nutritional status of patients with IBD is crucial for risk-stratifying.

Given the variable impact on clinical outcomes and the paucity of data regarding the prevalence of sarcopenia in Caucasians, we aimed to assess the impact of sarcopenia on clinical outcomes in a cohort of Caucasian patients with active CD undergoing CT-enterography (CTE) for clinical assessment in an emergency setting. We further investigated the prevalence of sarcopenia and its predictors.

## 2. Materials and Methods

This retrospective cohort study was conducted at an IBD tertiary academic center at the University of Naples Federico II and included adult IBD patients. Based on electronic medical records, we identified all CD patients between 2005 and 2018 who underwent CTE performed in an emergency setting. Baseline characteristics included demographic information, clinical, endoscopic, radiographic, and serologic data, and past and current medications. Clinical outcomes investigated in the current study were the rate of surgery and infections within 12 months of CTE.

### 2.1. Sarcopenia, Nutritional Status, and Body Composition Assessment

The skeletal muscle index (SMI) (cm^2^/m^2^), which is the ratio of the cross-sectional area of skeletal muscles at the level of the L3 vertebra to the height squared (m^2^), was used to detect sarcopenia in the early stages with a SMI < 38.5 cm^2^/m^2^ in women and <52.4 cm^2^/m^2^ in men [25,26].

BMI was calculated as weight (Kg) divided by height squared (m^2^), and malnutrition was diagnosed according to Global Leadership Initiative on Malnutrition (GLIM) criteria [27] with at least one phenotypic criterion including weight loss or reduced BMI and one etiologic criteria including reduced food intake/assimilation and disease burden/inflammation. Visceral obesity was defined as a visceral fat area (VFA) ≥130 cm [26].

### 2.2. CT Protocol

All exams were performed on a single CT scanner (Aquilion 64, Toshiba, Tokyo Japan). Each patient received 1200–1400 ml of a 7% polyethylene-glycol (PEG) solution orally administered from 45 to 60 min before CT examination, followed by ingestion of 500 mL of tap water immediately prior to scan acquisition. The scanning range was from the pubic symphysis to the diaphragm. The technical parameters were the following: detector configuration, 1 × 32 mm; rotation time, 0.75 s; tube voltage, 120 kVp; slice thickness, 5.0 mm; image reconstruction; filter (Kernel 3); scan direction, outward. All patients underwent a single pass (SP) contrast-enhanced abdominal protocol with a scan delay tailored around a monophasic contrast injection of 1.77 cc/kg of a non-ionic iodinated contrast media (370 mgI/mL), as previously reported [28].

### 2.3. Image Analysis

CT images were reviewed by a radiologist with 6 years of experience in abdominal imaging using the open-source Horos software (version 3.3.6). All measurements were performed on a single slice at the level of L3, with both transverse processes visible (Figure 1).

Muscle area (MA) was calculated as the cross-sectional area of all skeletal muscles, including the psoas, paraspinal muscles, and abdominal wall muscles, selecting an attenuation threshold ranging from −29 to 150 Hounsfield Unit (HU) [29]. Furthermore, skeletal mass index (SMI, MA/m2 body height) was obtained in order to correct for patients’ body height. Visceral and subcutaneous fat areas (VFA and SFA) were then calculated, setting an attenuation threshold between −190 and −30 HU, and VFA/SFA ratio was collected [30].

### 2.4. Statistical Analysis

Baseline descriptive statistics were reported to assess demographic and clinical characteristics and prevalence of sarcopenia in IBD patients. Continuous data were reported as mean ±standard deviation, and categorical data as a percentage.

Univariate analysis was used to evaluate correlations between demographic and clinical characteristics and sarcopenia. The *t*-tests were used to compare continuous variables, and chi-square or Fisher’s exact tests were used to compare categorical variables. A model to predict the presence of sarcopenia was realized using continuous logistic regression (multivariate analysis) with stepwise selection criteria on significant parameters. The logistic regression model is defined as:(1)$$P\left(x\right)=\frac{1}{1+e^{-g\left(x\right)}}$$
with
(2)$$g\left(x\right)=\beta_{0}+\beta_{1}x_{1}+\beta_{2}x_{2}+\cdots \ldots +\beta_{n}x_{n}$$
where *x*_1_, *x*_2_,… *x_n_* represent different input variables and β_0_, β_1__… _ β_n_ are the corresponding regression coefficients.

Model predictive power is quantified with the area under the receiver operating characteristic curve (AUC). Youden’s index was used to identify the best cut-off value that maximizes sensitivity and specificity. A level of significance was designated as a *p*-value < 0.05.

## 3. Results

### 3.1. Baseline Characteristics

A total of 63 patients with CD were identified and included in the analysis. Mean age was 44.2 years (range: 18–82), and males comprised 54% (34) of the study population. All patients had moderate–severe clinical activity of inflammation assessed by the Harvey–Bradshaw Index ≥ 5 [31] and a mean duration of disease of 140 ± 106 months. Most subjects (73%) had an ileo-colonic localization, with stricturing and penetrating behavior being observed in 57.1% and 39.7% of patients, respectively. Only 3.2% of subjects showed inflammatory behavior. A total of 13 (20.6%) patients reported extra-intestinal manifestations (EIM) such as cutaneous, ocular, and musculoskeletal. At the time of CTE assessment, 18 patients (28.6%) were treated with anti-TNF-α, 15 (23.8%) with steroids, and 14 (22.2%) with immunosuppressors. Demographic details are summarized in Table 1.

Based on the adopted SMI cut-off, 48 patients (68.3%) were sarcopenic without any significant difference between females and males (*p* = 0.33).

### 3.2. Clinical Outcomes

The overall rate of surgery was 33.0%, without any significant difference between sarcopenic and non-sarcopenic patients (*p* = 0.4). The rate of infections in patients with sarcopenia was significantly higher compared to the non-sarcopenic group (41.95% vs. 15%, *p* = 0.03). The most common reported infection was related to the peripherally inserted central venous catheter (PICC) (18.6%) (Table 2).

### 3.3. Assessment of Body Composition Measures and Association with Clinical Parameters

The BMI of patients with sarcopenia was lower than that of the non-sarcopenic ones (20.3 ± 3.1 kg/m^2^ vs. 23.3 ± 3.8 kg/m^2^, *p* = 0.002). A fair to moderate correlation was found between SMI and BMI (r = 0.4). A lower subcutaneous fat area (SFA) was observed in patients with sarcopenia compared to those without it (94.1 ± 90.4 cm^2^ vs. 149.9 ± 84.2 cm^2^, *p* = 0.009). Moreover, all subjects did not differ for VFA regardless of the presence or absence of sarcopenia, respectively (54.0 ± 63.1 cm^2^ vs. 63.4 ± 62.7 cm^2^, *p* = 0.58). The ratio VFA/SFA was higher in patients with sarcopenia compared with those without (0.7 ± 0.6 vs. 0.4 ± 0.4, *p* = 0.04).

There were no significant differences in C reactive protein (CRP), endoscopic activity measured with Simple Endoscopic Score for CD (SES-CD), and bowel wall thickness (BWT) assessed with ultrasound bowel sonography in the two cohorts (*p* > 0.05). Low albumin (3.5 ± 0.6 g/dL) and high fecal calprotectin (FC) (334.7 ± 167.2 μg/g) were observed in the presence of sarcopenia compared to 3.88 ± 0.5 (*p* = 0.029) and 238.95 ± 138.7 (*p* = 0.03) in patients without sarcopenia, respectively. EIM was associated with the presence of sarcopenia (*p* = 0.04). With regard to medical treatment, no differences were observed for biologic treatment, steroids, and immunosuppressors among sarcopenic and non-sarcopenic (*p* > 0.05).

Among patients treated with biologics (28.6%), 6 individuals (33.3%) reported infections without any significant association with anti-TNF-α (*p* = 0.76).

### 3.4. Predictors of Sarcopenia

In the entire patient cohort, BMI (OR 0.734, 95%CI 0.578–0.932) and EIM (OR 19.233, 95%CI 1.059–349.139) were the variables identified by logistic regression to predict sarcopenia (*p* < 0.05) (Table 3).

When they both were inserted in a prediction model, the ROC curve analysis resulted in an area under the ROC curve (AUROC) of 0.755 (95%CI 0.652–0.871) with a cut-off value of 0.53 (sensitivity = 88.4%, 95% CI 74.9–96.1%; specificity = 60.0%, 95%CI 36.1–80.9%; Youden Index = 0.48) (Figure 2).

We further calculate a model of the probability of the presence of sarcopenia based on the BMI and the presence/absence of EIM (Appendix A).

## 4. Discussion

During the last decade, short- and long-term treatment goals in IBD evolved, shifting from clinical remission to patient-centered parameters [32,33,34]. Additionally, there is increasing interest from leading IBD experts in the role of nutritional status in order to prevent the risk of malnutrition, re-establishing a good quality of life, and avoiding disability [35]. Furthermore, a comprehensive assessment of body composition may contribute to patient profiling. In this context, sarcopenia has become the focus of intense research aiming to translate recent knowledge of basic science in the field of intestinal immune system, cell trafficking and microbiota into improved early detection, intervention, and prevention of poor clinical outcomes.

Previous studies described the variable prevalence of sarcopenia ranging from 26 to 60%, with higher rates during relapse [16]. Hence, this highlights the need for a more homogenous and formal assessment of sarcopenia. In a recent multicenter retrospective cohort study comparing the prevalence of sarcopenia in IBD patients starting biologic therapy against patients undergoing surgery, the sarcopenia rate was 32% in the surgery cohort vs. 16% in the medical cohort [21]. Boparai et al. evaluated the prevalence of sarcopenia in 44 Asian patients affected by CD, reporting a muscle failure in almost 50% of them, more in females, regardless of age, disease severity, behavior, and location of disease [22]. In a further retrospective study, including 72 active CD patients hospitalized for relapsing disease, sarcopenia was observed in 42% of the population [5]. While in 344 patients with IBD in clinical remission, Unal found a malnutrition rate of 9.9%, a risk of malnutrition in 39.5% of patients, and sarcopenia and probable sarcopenia were diagnosed in 41.3% of cases [36].

In our cohort, more than two-thirds of patients were sarcopenic, while 44.4% were malnourished (among them, 85.7% were sarcopenic), and a significant correlation emerged between BMI and sarcopenia.

The impact of sarcopenia on clinical outcomes has been the objective of growing studies [16]. Evidence shows that sarcopenia is a risk factor for negative outcomes in IBD, including abscesses, hospitalizations, and digestive surgery in respect to non-sarcopenic patients [5,16,21,22,23,24]. However, our results showed that the presence of sarcopenia was not associated with the rate of surgery, endoscopic, and transmural activity assessed with ultrasound. Since our cohort included active CD who underwent CTE in an emergency setting, we believe that the activity itself is associated with the risk of surgery regardless of the presence of sarcopenia.

Importantly, there is already consolidated evidence that sarcopenia, as well as frailty, could be associated with an increased risk of infections [37,38], although there is a paucity of evidence in IBD. For instance, sarcopenic patients with type 2 diabetes showed a higher risk of infections compared with the non-sarcopenic group (OR = 1.469, 95% CI 1.102–2.031), particularly respiratory infections [38]. Similarly, among patients undergoing liver transplantation, muscle wasting was associated with an increased risk for post-transplant infectious complications and mortality (OR = 4.60, 95% CI 2.25–9.53) [39] and an independent preoperative predictor of infection after hepato-biliary-pancreatic surgery [40]. Of note, in the current study, we found that in active CD patients, sarcopenia was associated with infections, especially those related to PICC infection.

Consistent with previous studies [36], we confirmed the correlation of sarcopenia with malnutrition as a result of chronic and active inflammation leading to disease progression and, subsequently, more susceptibility to infections.

Furthermore, we analyzed VFA, SFA, and VFA/SFA ratios, observing a higher VFA/SFA ratio in sarcopenic patients than in non-sarcopenic ones. These results indicate that human fat tissue should be not only considered a reservoir for excess nutrients but a dynamic tissue involved in the regulation of immunity, inflammation, and angiogenesis [41]. Indeed, the VFA/SFA ratio and fibrofatty proliferation score play key roles in developing fibrostenosis in CD [42,43,44,45].

Noteworthy, we found a significant correlation with EIM. Since EIM frequently occurs in patients with IBD ranging from 19% to 40% and makes IBD “difficult to treat”, the assessment of nutritional status and body composition in addition to EIM should be investigated in routine clinical settings even in the case of remission and included in the treatment decision-making process [46,47,48,49].

Accordingly, we created a probability model for predicting sarcopenia, and we found that low BMI and the presence of EIM were associated with an increased probability of sarcopenia. A vast body of research has established a link between the immune system and/or microbiota of the gut and EIM [50,51]. Since the recent hypothesis of a “gut–muscle axis” [10], we speculate that cell trafficking pathways could be involved in transmitting signals from the gut to the muscle. Although joint and muscle axes are separate entities, they could be closely related due to cell migration, dysregulation of gut microbiota, and immune system. Hence these represent an important yet insufficiently explored field for potential future research.

With regards to IBD treatment, at the time of sarcopenia assessment, about one-third of patients were treated with anti-TNF-α. It has been described that biologic drugs, including anti-TNF-α agents, might decrease muscle failure, blocking the NF-kB signaling pathway and leading to a catabolic effect on skeletal muscle [12]. Subramaniam et al. showed that treatment with infliximab could reverse IBD-related sarcopenia in CD patients with active disease, improving both skeletal muscle volume and strength after a 6-month treatment [52]. Nevertheless, our findings did not show a significant difference between the anti-TNF-α group compared with the non-anti-TNF-α group in terms of sarcopenia.

In addition, since multiple studies [53] have demonstrated that patients receiving biologics are at high risk of infections, we investigated the infection rate associated with TNF-α inhibitors. However, in our cohort, there was no effect of TNF-α inhibitors on infections.

One strength of this study is that this is one of the few studies including the Caucasian population. In addition, sarcopenia has been extensively described in patients with IBD in clinical remission [21,54] though no studies have investigated sarcopenia in active CD. These point out that the assessment of sarcopenia in relation to subtype and disease activity is crucial to identifying diagnostic and management strategies with the final goal of improving disease outcomes. Moreover, to our knowledge, no studies have evaluated the association between infections, EIM, and sarcopenia in patients with IBD.

Conversely, this study had several limitations that should be acknowledged. First, the sample size was too small to provide an accurate estimate of sarcopenia prevalence in Caucasians and to detect significant differences in terms of surgical outcomes. The retrospective, single-center nature of this study represents a further limitation since we cannot provide a comprehensive assessment of sarcopenia. Furthermore, our cohort consisted of patients who underwent CT scans in an emergency setting due to active CD, so at the moment of the exam, we could not include measures of muscle strength and function, which are traditionally required for the full definition of sarcopenia.

However, Sarcopenia: revised European consensus on definition and diagnosis [14] highlights that high-resolution imaging such as CT scan is a reliable tool to detect sarcopenia in early stages and in specialty care for individuals at high risk of adverse outcomes such as our cohort of active CD patients.

Regarding image analysis, interobserver agreement was not assessed; nevertheless, the investigator responsible for the task is highly experienced in image segmentation [55,56].

## 5. Conclusions

In conclusion, sarcopenic patients with active CD have a higher rate of infections, suggesting a potential role of sarcopenia as a useful measure to identify patients who are at higher risk of adverse clinical outcomes.

Given the higher rate of sarcopenia in patients with EIM and malnutrition, assessing IBD patients for sarcopenia and nutritional and body composition status are increasingly recognized as important for risk-stratifying patients and implementing a dietary intervention. Areas for future studies should focus on investigating the most appropriate threshold for defining sarcopenia and validating it in a larger IBD Caucasian population. Furthermore, we hope to encourage the assessment of sarcopenia through tools that are easy to use in practice, cost-effective, and applicable in several clinical settings. Finally, prospective studies are needed to determine the effect of early treating sarcopenia on clinical outcomes and disability.

## Figures and Tables

**Figure 1 nutrients-14-03460-f001:**
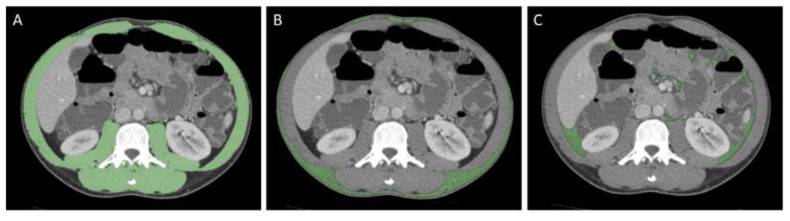
Representative example of assessment of muscle area (**A**), subcutaneous (**B**), and visceral fat areas (**C**) on a single slice CT-enterography at the level of L3 in a 25 yo male patient.

**Figure 2 nutrients-14-03460-f002:**
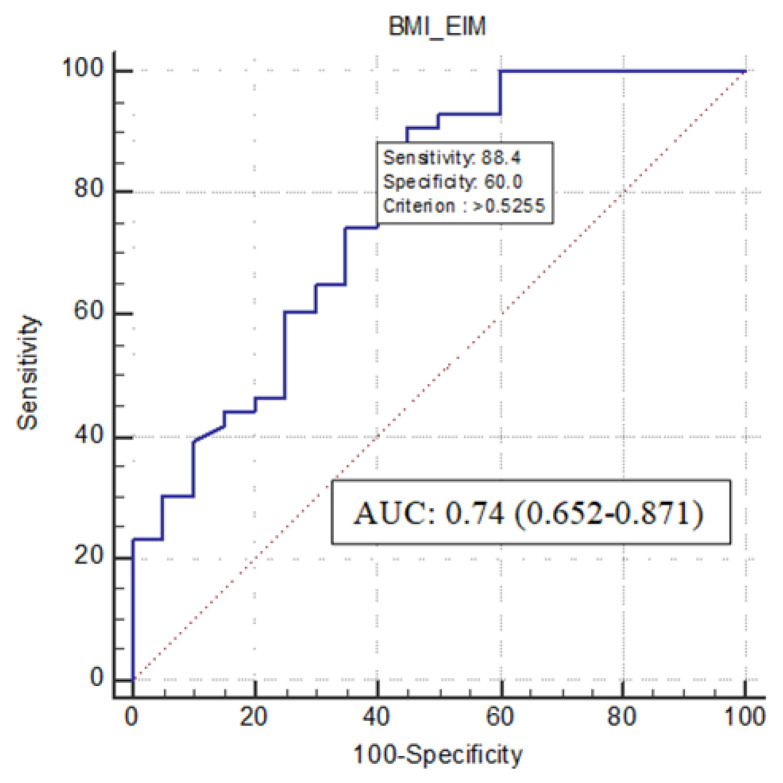
Diagnostic accuracy of BMI and EIM for predicting sarcopenia. Receiver Operating Characteristic curve to evaluate the sensitivity and specificity of the model to predict the presence of sarcopenia.

**Table 1 nutrients-14-03460-t001:** Baseline demographic characteristics of cohort.

	*Total (n = 63)*
*Age*	44.2 ± 17.0
*Sex*	
* * *Male*	34 (54.0)
* * *Female*	29 (46.0)
*BMI (kg/m^2^)*	21.2 ± 3.6
*SMI (cm^2^/m^2^)*	43.6 ± 9.1
*SFA (cm^2^)*	111.8 (91.6)
*Sarcopenic patients*	43 (68.3)
*VFA (cm^2^)*	57.0 ± 35.0
*VFA/SFA*	0.62 ± 0.58
*Disease duration (months)*	140.0 ± 106.5
*CD Montreal disease localization*	
* * *Ileal*	13 (20.6)
* * *Colonic*	3 (4.8)
* * *Ileo-colonic*	46 (73.0)
* * *Isolated upper-GI*	1 (1.6)
*CD Montreal disease behavior*	
* * *Inflammatory*	2 (3.2)
* * *Stricturing*	36 (57.1)
* * *Penetrating*	25 (39.7)
*Perianal disease*	15 (23.8)
*HBI*	7.8 ± 1.7
*SES-CD*	7.8 ± 5.9
*BWT (mm)*	7.0 ± 2.2
*CDE (cm)*	17.8 ± 14.6
*Extraintestinal manifestations*	13 (20.6)
* * *Musculoskeletal manifestations*	11 (17.5)
* * *Cutaneous manifestations*	1 (1.6)
* * *Ocular Manifestations*	1 (1.6)
*Baseline anti-TNF-α therapy*	18 (28.6)
*Baseline CCS therapy*	15 (23.8)
*Baseline ISS therapy*	14 (22.2)
*Previous anti-TNF-α therapy*	26 (41.3)
*CRP (mg/L)*	2.9 ± 4.0
*FC (μg/g)*	302.7 ± 163.4
*Serum albumin (g/dL)*	3.6 ± 0.6

Values are mean ± SD or *n* (%); BMI, Body mass index; SMI, Skeletal muscle index; SFA, Subcutaneous fat area; VFA, Visceral fat area; CD, Crohn’s disease; SES-CD, Simple Endoscopic Score for Crohn’s Disease; HBI, Harvey Bradshaw Index; SES-CD, Simple endoscopic score for Crohn’s disease; BWT, bowel wall thickness; CDE, Crohn’s disease extent; CCS, Systemic corticosteroids; ISS, Traditional immunosuppressive therapy; CRP, C-reactive protein; FC, Fecal calprotectin.

**Table 2 nutrients-14-03460-t002:** Univariate analysis on anthropometry, patients’ characteristics and outcomes in the sarcopenic and non-sarcopenic cohort.

	Patients with Sarcopenia (n = 43)	Patients without Sarcopenia (n = 20)	*p*-Value
BMI (kg/m^2^)	20.3 ± 3.1	23.3 ± 3.8	0.002
SFA (cm^2^)	94.1 ± 90.4	149.9 ± 84.2	0.009
VFA (cm^2^)	54.0 ± 63.1	63.4 ± 62.7	0.54
VFA/SFA ratio	0.7 ± 0.6	0.4 ± 0.4	0.04
Age	45.6 ± 18.0	41.1 ± 14.8	0.49
CRP (mg/L)	2.91 ± 4.1	2.98 ± 3.8	0.54
Serum albumin (g/L)	3.49 ± 0.6	3.88 ± 0.5	0.029
FC (μg/g)	334.66 ± 167.2	238.95 ± 138.7	0.03
Female gender	18 (41.7)	11 (55.0)	0.33
EIM	12 (27.9)	1 (5.0)	0.04
Baseline anti-TNF-α therapy	10 (23.2)	8 (40.0)	0.17
Baseline CCS therapy	8 (18.6)	7 (35.0)	0.15
Baseline ISS therapy	9 (20.9)	5 (25.0)	0.72
Previous anti-TNF-α therapy	20 (46.5)	6 (30.0)	0.21
Need for surgery	21 (48.8)	12 (60.0)	0.41
Infectious events	18 (41.9)	3 (15.0)	0.03
Low respiratory tract	6 (13.9)	1 (5.0)	
PICC-related	8 (18.6)	0	
Urinary tract	1 (2.3)	1 (5.0)	
Herpesvirus-related	2 (4.6)	1 (5.0)	
Gynaecologic	1 (2.3)	0	

Values are mean ± SD or *n* (%); BMI, Body mass index; SFA, Subcutaneous fat area; VFA, Visceral fat area; VFA/SFA, CRP, C-reactive protein; FC, fecal calprotectin; EIM, extra-intestinal manifestations; CCS, Systemic corticosteroids; ISS, Traditional immunosuppressive therapy; PICC, peripherally inserted central catheter.

**Table 3 nutrients-14-03460-t003:** Multivariate analysis: logistic regression coefficients and 95% confidence intervals.

					95% C.I. per EXP(B)	
	Estimated Coefficient	Standard Error	*p*-Value	Exp(B)	Lower Limit	Upper Limit
Body Mass Index	−0.309	0.122	0.011	0.734	0.578	0.932
EIM	2.957	1.479	0.046	19.233	1.059	349.139
Constant	6.855	2.591	0.008	948.241		

EIM, extraintestinal manifestations; CI, confidence interval.

## Data Availability

Data will be shared on reasonable request to the corresponding author with the permission of all co-authors.

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
