# Peer review of "Impact of Sarcopenia on Clinical Outcomes in a Cohort of Caucasian Active Crohn’s Disease Patients Undergoing Multidetector CT-Enterography"

_nutrients, 2022, doi:10.3390/nu14173460_

Round 1
Reviewer 1 Report
Thank you for inviting me to review this manuscript.
I have minor comments only. The manuscript is well written and planned well.
The authors have identified a key area to address in IBD patients.The methodology has been explained clearly. Although the number of patients are not very high, the results are interesting and perhaps applicable to this cohort of patients.
The correlation with EIMs is interesting and requires further research. I suggest the authors make this clear and if possible put forward a hypothesis or explanation.
Author Response
We thank the reviewer for the positive and encouraging comments.
We have now included a possible explanation on this in the discussion page 8 line 255-261
“A vast body of research has established a link between the immune system and/or microbiota of the gut and extra-intestinal manifestations [Zundler S et al, Nature Rev 2022 ; Schett G. et al, N. Engl. J. Med. 2021]. Since the recent hypothesis of a “gut-muscle axis” [Nardone OM, Frontiers Immunol 2021] we speculate that cell trafficking pathways could be involved in transmitting signals from the gut to muscle. Although joint and muscle axis are separate entities, they could be closely related due to cell migration and dysregulation of gut microbiota and immune system. Hence these could represent an important yet insufficiently explored field for potential future research.

Reviewer 2 Report
The manuscript entitled „Sarcopenia and altered body composition are associated to increased risk of infection in active Crohn’s disease patients” presents interesting issue, but some problems should be corrected. In the present form this material presents rather draft of the manuscript, than manuscript of publication ready to be reviewed and published.
1. The major problem associated with presented manuscript results from the fact that it is shabbily prepared and the parts of the study do not correspond each other. The most important issue may be indicated for the major parts of the study: title (assessment of the risk of infection) does not correspond aim of the study (sarcopenia frequency) and does not correspond results (diagnostic accuracy of BMI and EIMs), as each part indicates different scope. As a result reader do not know what was the main idea of Authors for this study – what did they want to present here.
Taking this into account, the major correction is necessary and more scientific discipline is needed – Authors must ask themselves a most important question about what is the real aim of the study, and afterwards adjust the other parts. Authors must be aware that the title must be associated with the aim, and result must be within the aim of the study. The other parts must be corrected as well, as Introduction must justify the aim, and Discussion must present the results of the other studies within this aim.
2. Authors assessed a relatively small sample of 63 patients of very diverse characteristics (age 18-82, females and males combined, treated with various therapeutic options), which causes that the possibility to conclude is limited. For such heterogenic group Authors must carefully formulate their aim.
Authors declared that 18 patients (28.6%) were treated with anti-TNF-alpha – I suppose that it was Infliximab or a similar medication. This is a problem, as patients receiving TNF-α inhibitors are at high risk of infections, so formulating a statement about the risk of infection within such a group must be based on deepen statistical analysis including applied pharmacotherapy. Authors must get familiar with the current state of knowledge including anti-TNF-alpha medications and their influence (e.g. https://pubmed.ncbi.nlm.nih.gov/25630559/)
3. Last but not least, Authors indicate that they diagnosed sarcopenia, but they did not diagnosed it based on proper diagnostic criteria. Authors should get familiar with the current recommendations within clinical algorithm used for sarcopenia (https://www.ncbi.nlm.nih.gov/pmc/articles/PMC6322506/), which indicates that 3 elements should be assessed to diagnose sarcopenia: muscle strength, muscle quantity and physical performance, as without 3 elements combined we are unable to diagnose sarcopenia properly. Authors did not assess 3 required elements, but only one of them (muscle quantity – “The skeletal muscle index (SMI) was used to assess sarcopenia”), so they can not state that they diagnosed sarcopenia – in fact they are unable to diagnose sarcopenia based on this element only.
Round 2
Reviewer 2 Report
The manuscript entitled „Impact of sarcopenia on clinical outcomes in a cohort of Caucasian active Crohn’s disease patients undergoing multidetector CT-enterography” presents interesting issue, but some problems should be corrected. In the present form this material presents rather draft of the manuscript, than manuscript of publication ready to be reviewed and published. In spite of the fact that Authors included some corrections, they still did not polish manusctipt adequately.
The major problem associated with presented manuscript results from the fact that it is shabbily prepared and the parts of the study do not correspond each other. The most important issue may be indicated for the major parts of the study: title (impact od sarcopenia) does not correspond aim of the study (sarcopenia frequency) and does not correspond results (diagnostic accuracy of BMI and EIMs), as each part indicates different scope. As a result reader do not know what was the main idea of Authors for this study – what did they want to present here.
Taking this into account, the major correction is necessary and more scientific discipline is needed – Authors must ask themselves a most important question about what is the real aim of the study, and afterwards adjust the other parts. Authors must be aware that the title must be associated with the aim, and result must be within the aim of the study. The other parts must be corrected as well, as Introduction must justify the aim, and Discussion must present the results of the other studies within this aim.
Authors indicate that they diagnosed sarcopenia, but they did not diagnosed it based on proper diagnostic criteria. Authors should get familiar with the current recommendations within clinical algorithm used for sarcopenia (https://www.ncbi.nlm.nih.gov/pmc/articles/PMC6322506/), which indicates that 3 elements should be assessed to diagnose sarcopenia: muscle strength, muscle quantity and physical performance, as without 3 elements combined we are unable to diagnose sarcopenia properly. Authors did not assess 3 required elements, but only one of them (muscle quantity – “The skeletal muscle index (SMI) was used to assess sarcopenia”), so they can not state that they diagnosed sarcopenia – in fact they are unable to diagnose sarcopenia based on this element only. Authors should rather refer the assessed as sarcopenia risk or just muscle mass (being associated with sarcopenia) - it would be the most accurate.
